# Race-based biases in psychological distress and treatment judgments

**Jonathan W. Kunstman** [1][☯]*, **Tade Ogungbadero**[2][☯], **Jason C. Deska**[3], **Michael J. Bernstein**[4], **April R. Smith**[1], **Kurt Hugenberg**[5]

1 Department of Psychological Sciences, Auburn University, Auburn, Alabama, United States of America,
2 Fisher College of Business, The Ohio State University, Columbus, OH, United States of America,
3 Department of Psychology, Toronto Metropolitan University, Toronto, Canada, 4 Psychological and Social Sciences, Pennsylvania State University–Abington, Abington, PA, United States of America, 5 Department of Psychological and Brain Sciences, Indiana University, Bloomington, IN, United States of America

☯ These authors contributed equally to this work.
* jwk0030@auburn.edu

**Data Availability Statement:** All data files are available on the Open Science Framework. https://osf.io/kj6yz/?view_only=a201a27d263e42a4af60d4f563601cb8.

## Abstract

Racism creates and sustains mental health disparities between Black and White Americans and the COVID-19 pandemic and ongoing harassment directed at Black Americans has exacerbated these inequities. Yet, as the mental health needs of Black Americans rise, there is reason to believe the public paradoxically believes that psychopathology hurts Black individuals *less* than White individuals and these biased distress judgments affect beliefs about treatment needs. Four studies (two pre-registered) with participants from the American public and the field of mental health support this hypothesis. When presented with identical mental illnesses (e.g., depression, anxiety, schizophrenia), both laypeople and clinicians believed that psychopathology would be less distressing to Black relative to White individuals. These distress biases mediate downstream treatment judgments. Across numerous contexts, racially-biased judgments of psychological distress may negatively affect mental healthcare and social support for Black Americans.

## Introduction

Interpersonal and institutional racism take an unconscionable toll on the mental health of Black Americans, and existing health disparities have been magnified by the strain of the COVID-19 pandemic and ongoing epidemics of police violence and public intimidation directed at Black Americans [1–5]. Consequently, Black Americans' mental health suffers at significantly greater rates than White Americans [6–8]. Yet, despite *greater* harms to their mental health, people may paradoxically believe psychopathology harms Black Americans *less* than White Americans. People frequently believe that past hardships desensitize Black individuals to everyday social hurts [9, 10]. The current work tested whether racial biases in social pain judgments extend to beliefs about the harms of psychopathology. We predicted that when presented with identical mental illnesses (e.g., depression, anxiety, schizophrenia), participants would believe that psychopathology was less distressing to Black relative to White individuals. Moreover, we predicted that these distress biases would mediate effects on expected treatment

**Funding:** The authors received no specific funding for this work.

**Competing interests:** The authors have declared that no competing interests exist.

needs. Four studies (two pre-registered) tested these hypotheses among the lay public (Studies 1-2b) and mental health professionals (MHPs; Studies 1,3).

## Race and distress

Converging evidence supports the hypothesis that people will expect psychopathology to hurt Black individuals less than White individuals. First, evaluators often deny Black individuals complex emotions like grief, suffering, and mental anguish [11]. Second, parallel research on the socio-emotional experiences of Black individuals finds that Black Americans regularly feel their social pains are invalidated and minimized [12, 13]. Third, research documents racial biases in pain judgments such that people regularly believe the same hurtful events (e.g., breaking a bone; being disrespected) harm Black individuals less than White individuals [10, 14]. As this work attests, racial biases frequently result in the minimization of Black individuals' pain.

These racial biases in pain judgments are multiply determined. Cross-race deficits in communication, facial processing, social accuracy, biological racism, and beliefs about the adversity and toughness of Black individuals have all been linked to biased judgments of physical pain [15–20]. Like physical pain, racialized beliefs about hardship and toughness also bias judgments of social pain (i.e., distress and negative emotions related to hurtful experiences with others [9, 10]). Subsequent research finds that factors associated with the activation of Black racial stereotypes (i.e., Black racial phenotypicality [21]) modulated these effects, such that individuals with more phenotypically Black features had their pain minimized to a greater degree than individuals with less phenotypically Black features [9]. In both cases, racial biases in social pain were driven by beliefs about the toughness and adversity experienced by Black individuals. People believe Black individuals experienced considerable adversity and consequently were desensitized to everyday social pains. When viewed collectively, the above research provides consistent evidence for predicting that race will bias judgments of psychological distress, such that mental illness is believed to hurt Black individuals less than White individuals.

## Contribution of the current work

The current work makes several contributions to research on pain, race, and mental health. First, these studies identify a racial bias in distress and treatment judgments relevant to the mental health crisis facing Black Americans. Over 400 years of racial violence and resultant social and economic inequality has produced sustained mental health disparities between Black and White Americans [22–25]. Yet, precisely when Black Americans mental health needs are pressing, many in society may paradoxically believe that Black individuals are hurt less and have weaker treatment needs than White individuals. The current work stands to provide evidence for a social-cognitive bias that undermines judgments of the mental health needs of Black Americans.

Second, the minimization of Black individuals' psychological distress hints at a novel theoretical mechanism that may affect mental healthcare for Black clients. Heretofore, research on racial biases in mental health have focused on (mis)communication, trust, symptom interpretation, access to health services, and the role of prejudice in diagnosis and treatment [15, 26–28]. Racial biases in distress judgments complement this research and—because evaluations of distress are critical to every stage of mental healthcare [29]—provide evidence for a bias with wide-ranging treatment implications.

Finally, the current work stands to advance research on pain biases beyond judgments of common social hurts to beliefs about the nature of psychopathology and mental healthcare. Rather than being limited to discrete and moderate social stressors (e.g., disrespect, exclusion

[10]), target race stands to bias judgments of chronic and extreme mental health conditions that shape the daily lives of countless Americans. Race-based biases in pain judgments extend to fundamental judgments of psychological distress and treatment needs.

## The current work

Four studies tested whether people believe psychopathology harms Black individuals less than White individuals and whether biased distress judgments mediate racial biases in judgments of treatment needs. Study 1 paired images of Black and White individuals with ten mental illnesses and tested whether participants believed these illnesses would cause Black individuals less distress than White individuals. Study 1 also preliminarily tested whether MHPs (e.g., counselors, clinical psychologists, psychiatrists) showed similar racial biases in distress judgments. Study 2a-2b (pre-registered replication) tested whether racial biases in distress mediated effects on treatment judgments. Study 3 (pre-registered) tested whether racial biases in distress and treatment judgments extend to practicing MHPs.

## Study 1: Race-based bias in mental health distress judgments

The current study's primary goal was to test our main hypothesis related to racial biases in psychological distress judgments. To test this hypothesis, ten Black and ten White male targets from the Chicago Face Database (CFD [30]) were paired with ten forms of psychopathology. The CFD is a frequently used and well validated stimulus set of adult target individuals from a variety of racial and ethnic backgrounds. Using norming data associated with face stimuli, we identified Black and White target individuals whose racial identity was easily identifiable while holding constant other facial features that might affect distress judgments (e.g., dominance, attractiveness, baby-facedness). Thus, using normed CFD stimuli allowed us to manipulate target race without introducing confounds related to facial structure and morphology. Participants then judged the distress they believed each mental illness would cause these pictured individuals. We hypothesized that participants would expect psychopathology to harm Black individuals less than White individuals.

The study's secondary goal was to explore whether potential biases in psychological distress generalize to those working in the field of mental health. With this goal in mind, we included a demographic item that allowed professional Mental Healthcare Professionals (MHPs; e.g., counselors, clinical psychiatrists, psychiatrists) to self-identify. In light of evidence that implicit bias and stereotype endorsement is found among healthcare workers [17, 28, 31], there was good reason to predict racial biases in psychopathology judgments would extend to mental health professionals.

### Method

**Participants.** Ethics approval was granted for this research by Miami University and Auburn University's Internal Review Boards, respectively. Written consent was presented via computer.

Using the effect of target race documented in Deska, Kunstman et al. (2020b) [10] $d$ = .52, an a priori power analysis with G*Power software (V3.1 [32]) suggested a sample size of 32 participants would provide 80% power ($\alpha$ = .05) for the primary hypothesis test with a paired-samples t-test. In anticipation of data loss and to provide a more robust test, we oversampled and collected data from 204 U.S. participants via the online research participation platform MTurk. MTurk is an internet-based participant service that allows researchers to connect with individuals interested in participating in social science research in exchange for payment to Amazon Marketplace. The identification of duplicate IP addresses resulted in 11 exclusions, leaving an analyzable sample of 196 (1% American Indian or Alaska Native; 7% Asian; 19%

Black or African American; 3% Latina/o; 1% Multi-racial; White 66%; 34% female; 63% male; $M_{age}$ = 33.83, SD = 10.30). Including excluded participants does not change the direction or magnitude of any effects reported in Studies 1-2b. Differences related to analyzing complete data for Study 3 are discussed below. Full sample analyses for each study are located in the supplemental online materials. At no time did researchers have access to information that would reveal participants' identities. Sensitivity analyses indicated this sample provided power to detect minimum effects of d = .20. Data and materials for the complete work and pre-registration plans for Studies 2b and 3 can be located at (https://osf.io/kj6yz/?view_only=a201a27d263e42a4af60d4f563601cb8).

**Materials.** Participants viewed images of ten Black and ten White male targets from the Chicago Face Database (CFD [30]). Black and White targets were equated on ratings of age, anger, attractiveness, baby-facedness, dominance, femininity, happiness, masculinity, racial prototypicality, sadness, surprise, threat, trustworthiness $F_{range}$ = 0.01–2.28, $p_{range}$ = 0.92–0.14. Specific tests of CFD norming data are available in the online materials and via the OSF link above. Target individuals were paired with a ten-item measure assessing harm caused by ten forms of psychopathology: depression, social anxiety, arachnophobia, post-traumatic stress disorder (PTSD), bipolar disorder, panic disorder, insomnia, obsessive-compulsive disorder, generalized anxiety disorder, and schizophrenia. Participants rated how much psychological distress the pictured individual would experience if they had each form of psychopathology. Ratings were made on a 1 (*not painful*) to 7 (*extremely painful*) Likert scale. Distress judgments were then averaged within targets and combined across racial categories. Target images and psychopathology items were randomized.

Participants' background in mental health was assessed with a single item embedded within demographic questions. Specifically, participants were asked (yes/no) whether they provide professional mental health services to others (e.g., as a psychiatrist, clinical psychologist, professional counselor). Sixty-one participants responded yes to this item, indicating they had experience providing professional mental health services.

**Procedure.** After consenting, participants completed the twenty-trial distress judgments task in an independently randomized order. After completing demographic information, participants were thanked, debriefed, and compensated for their time.

## Results

A paired-samples t-test assessed the primary hypothesis that participants would believe psychopathology harms Black individuals less than White individuals. As hypothesized, when paired with the same forms of psychopathology, Black target individuals were judged to experience less distress (M = 4.52, SD = 1.30) than White target individuals (M = 4.64, SD = 1.26), t (192) = -3.963, p < .001, 95% CI[-.19, -.064], d = 0.29.

Since a meaningful subset of our 196 participants self-reported as MHPs (n = 61), this allowed us to perform an exploratory test whether responses were moderated by professional background. To investigate this hypothesis, we conducted a mixed-model ANOVA where background (MHP/lay public) was entered as a between-subjects factor and ratings of psychological distress for targets (Black/White) were entered as a within-subjects factor. Sensitivity analysis revealed this test had 80% power (α = .05) to detect small-to-medium effects (d = .40). Results of this analysis reproduced the previously described main effect of target race, $F(1, 190)$ = 7.85, p = .006, $\eta_p^2$ = .060 and produced a main effect of background, such that MHPs expected target individuals to experience more distress (M = 5.00, SD = 1.17) than members of the public (M = 4.39, SD = 1.26). The interaction between participant background and target race was not significant, $F(1, 190)$ = .36, p = .549, $\eta_p^2$ = .002.

## Discussion

These results provide initial evidence that the public believes psychopathology harms Black individuals less than White individuals. Moreover, although clinicians expected target individuals to experience more distress than those in the lay public, racial bias in distress judgments were largely comparable between MHPs ($d = .25$) and the lay public ($d = .30$). Although these results should be interpreted with caution given the relatively small sample of MHPs, there is some preliminary evidence that racial biases in psychological distress may extend to those working in mental health. In Studies 2a-2b, we next tested whether racially-biased distress judgments mediated effects on judgments of treatment needs.

## Study 2a-2b: Biased distress judgments inform treatment recommendations

Study 2a had two primary goals. First, we aimed to further test the central hypothesis that people believe psychopathology hurts Black individuals less than White individuals. Second, we aimed to build on the bias documented in Study 1 to test whether racial biases in distress inform judgments of treatment needs. To test this treatment hypothesis, we added items assessing how many sessions of therapy pictured individuals would need to cope with mental illness. We hypothesized that biases in distress would bias treatment recommendations, such that participants would believe Black target individuals needed less treatment than White target individuals. Study 2b served as a direct replication of Study 2a's effects.

## Method

**Participants Study 2a.**　Using Study 1's effect estimate ($d = .51$), a priori power analysis (80% power; $\alpha = .05$), suggested a sample of 33 participants would be needed to test the study's main hypothesis with a paired-samples $t$-test. We oversampled and collected data from an undergraduate student sample across a full academic term. This sampling strategy resulted in a sample of 198 participants. Of this sample, 29 participants failed at least one of the study's two attention checks, yielding an analyzable sample of 167 (9% Asian; 2% Black or African American; 2% Latina/o; 5% Multi-racial; White 78%; 55% female; 43% male; 2% did not disclose their sex; $M_{age} = 19.20$, $SD = 1.24$). Sensitivity analysis indicated that with 80% power ($\alpha = .05$) this sample could detect effects as small as $d = .22$.

**Participants 2b.**　As a pre-registered replication of Study 2a, we sought a minimum analyzable sample of 190 participants. Expecting incomplete responses and exclusion, we oversampled 278 participants from MTurk. Of this initial sample, 68 participants failed at least one of the study's attention checks or produced a duplicate IP address (suggesting repeat participation), leaving an analyzable sample of 210 (1% American Indian or Alaska Native; 5% Asian; 29% Black or African American; 3% Latina/o; 2% Multi-racial; 59% White; 39% female; 61% male; 1% chose not to disclose their sex; $M_{age} = 35.90$, $SD = 11.20$). Sensitivity analysis performed on this sample indicated that the current study had 80% power ($\alpha = .05$) to reliably detect effects as small as $d = .19$.

**Materials 2a and 2b.**　Participants completed the same distress task as participants in Study 1 with two noteworthy modifications. Participants were not asked about their experience providing professional mental healthcare. Participants also made treatment judgments for each mental health item. For example, following distress judgments of depression, participants were asked, "How many sessions of therapy do you believe this person would need to effectively cope with depression?" Participants made judgments on a 1 (*0 sessions*) to 27 (*26+ sessions*) Likert scale. Judgments of pain and treatment were averaged across targets and racial categories.

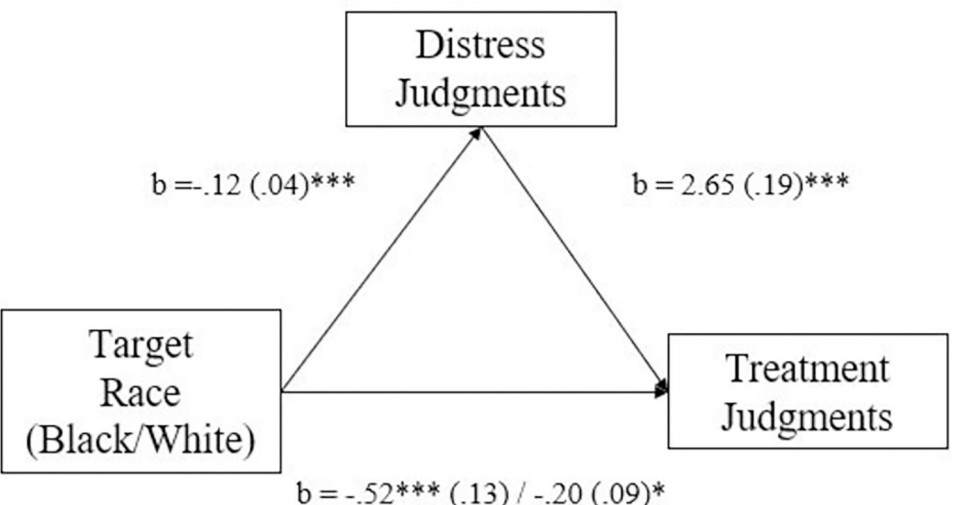

**Fig 1. Study 2a, effect of target race on treatment judgments is mediated by distress judgments.** $^* p \leq .05$, $^{**} p \leq .01$, $^{***} p \leq .001$.

*Procedure*. Following consent, participants completed the twenty-trial distress and treatment task. Trial order and psychopathology items were randomized. Participants then completed demographic information, were debriefed, and compensated for their time.

## Results 2a

Paired samples t-tests were performed to compare judgments of Black and White targets' psychological distress and treatment needs. Participants again judged Black targets ($M$ = 4.14, $SD$ = 1.04) to feel less psychological distress than White targets ($M$ = 4.32, $SD$ = .93), $t(166)$ = -6.38, $p < .001$, 95% CI[-.24, -.12], $d$ = .49. Participants also believed that compared to White targets ($M$ = 12.03, $SD$ = 4.71), Black targets ($M$ = 11.23, $SD$ = 4.92) would need less treatment to manage their mental health, $t(166)$ = -8.26, $p < .001$, 95% CI[-.99, -.61], $d$ = 0.64.

Using the MEMORE macro to test within-subjects mediation [33], we tested whether race's effect on treatment judgments was mediated by judgments of psychological distress (Fig 1). Results indicated the effect of target race on treatment judgments was mediated by racial biases in distress judgments, $b$ = -.32, $SE$ = .11, 95% CI[-.55, -.13].

## Results 2b

Paired-samples *t*-tested compared judgments of distress and treatment for Black and White targets. Participants again judged Black targets ($M$ = 4.47; $SD$ = 1.06) to experience less distress than White targets ($M$ = 4.54; $SD$ = 1.04), $t(209)$ = -2.59, $p$ = .01, 95% CI[-.12, -.02] $d$ = 0.18, and expected Black targets ($M$ = 10.87; $SD$ = 5.06) to need fewer treatment resources than White targets ($M$ = 11.18; $SD$ = 5.19), $t(209)$ = -3.18, $p$ = .002, 95% CI[-.50, -.12] $d$ = 0.22 (Fig 2).

We next tested whether judgments of psychological distress mediated race's effect on treatment judgments. Results indicated that target race's effect on treatment judgments was mediated by racial biases in psychological distress, $b$ = -.19, $SE$ = .07, 95% CI[-.34, -.05]. Participants believed Black targets would experience less psychological distress and consequently require fewer treatment resources than White targets.

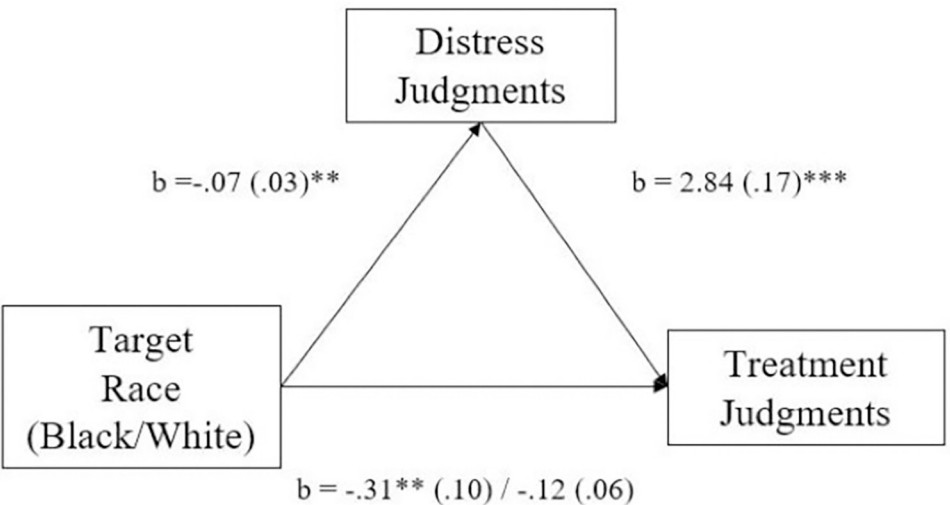

**Fig 2. Study 2b effect of target race on treatment judgments is mediated by distress judgments.** * $p \le .05$, ** $p \le .01$, *** $p \le .001$.

### Discussion 2a-2b

Participants believed Black individuals would experience less distress and need less psychological treatment than White individuals. Moreover, biases in distress mediated biased treatment judgments. Together, these results provide additional evidence for racial biases in beliefs about psychopathology and illustrate that these biases contaminate judgments of treatment needs. Participants believed that psychopathology harmed Black individuals less than White individuals and subsequently believed Black individuals needed less treatment relative to White individuals. Study 3 next tested whether racially-biased distress judgments extend to clinicians and mediate effects on treatment recommendations among practicing clinicians.

## Study 3: Tests among mental healthcare professionals

Having provided consistent evidence of racial biases in judgments of psychological distress and treatment needs among the public, the current study tested whether these effects generalized—or were moderated—by experience in the field of mental healthcare. Consequently, we collected data from mental health professionals (MHPs) and members of the lay public. In light of preliminary evidence that MHPs showed smaller-but-comparable racial biases in psychological distress as laypeople (Study 1), there was reason to suspect similar biases in treatment judgments, such that Black target individuals would be judged to need less treatment than White target individuals. To test these hypotheses, laypeople and MHPs completed the distress and treatment task from Studies 2a-b.

### Method

**Participants.** We recruited 160 mental healthcare professionals and 258 members of the general public. The size of the mental health professional portion of our sample was the largest possible given the resource costs needed to compensate this highly-skilled and difficult-to-reach population ($100.00 payments per clinician). MHPs were recruited through Qualtrics panel services by disseminating the survey to individuals who previously indicated that they provided mental healthcare as a counselor, clinical psychologist, psychiatrist, social worker, etc. At the start of the survey, participants indicated which field best represented their

profession. To be eligible to continue participating, individuals needed to select the response option labeled 'Mental Health: counselors, clinicians, therapists, psychologists, psychiatrists, social workers, etc.' Individuals indicating other professions were funneled out of the survey. Lay participants were recruited through CloudResearch. Like MTurk, Cloudresearch is an online participant platform that allows interested individuals to participate in research in exchange for monetary compensation. Among lay participants, 176 identified as White, 4 as American Indian or Alaska Native, 12 as Asian, 46 as Black or African American, 14 as Latina/o, 4 as Multi-racial, 1 participant identified as Hawaiian or Pacific Islander and 1 participant did not disclose their race (164 female; 93 male; 1 did not disclose; $M_{age}$ = 43.29, $SD$ = 17.61). Among MHPs, 112 identified as White, 4 as American Indian or Alaska Native, 23 as Asian, 6 as Black or African American, 9 as Latina/o, 3 as Multi-Racial, 3 participants did not disclose their race (41 female; 119 male; $M_{age}$ = 47.59, $SD$ = 13.99). Here we would note that although detailed demographics for MHPs were cut to conform to the contracted study duration with Qualtrics, soft launch data from the first 30 MHPs indicated participants held advanced healthcare degrees (e.g., Ph.D., M.D.) with at least five years of clinical practice (range = 5–36 years of clinical experience). Of the total sample of 418, 139 participants were excluded for failing the study's attention check or for presenting a duplicate IP address that could indicate bot-responding ($n_{public}$ = 100; $n_{MHP}$ = 39). After exclusions, the study achieved an analyzable sample of 262 ($n_{public}$ = 152; $n_{MHP}$ = 110). Sensitivity analysis suggests this sample provided 80% power ($\alpha$ = .05) to detect an interaction between participants' background and target race as small as $d$ = .34.

**Materials.** Using the task from Studies 2a-b, participants made judgments of psychological distress and treatment needs. However, to account for greater nuance in treatment recommendations, the treatment scale was expanded. Units 1–26 were on an interval scale 1 (*0 sessions*), 2 (*1 session*). . .26(*25 sessions*); options 27 (*26–52 sessions*) and 28 (*52+ sessions*) encompassed greater amounts of therapy sessions.

**Procedure.** Following consent, participants completed the distress and treatment task for Black and White targets. After completing demographic items, participants were debriefed, thanked, and compensated for their time.

## Results

We conducted a mixed-model ANOVA where participant background (public/MHP) was entered as a between-subjects factor and distress judgments corresponding to target race (Black/White) were entered as a within-subjects factor. Results of this analysis yielded a significant main effect of target race, $F(1, 260)$ = 12.39, $p$ = .001, $\eta_p^2$ = .045, such that Black targets ($M$ = 4.44, $SD$ = 1.43) were judged to experience less psychological distress than White targets ($M$ = 4.54, $SD$ = 1.39). Neither the main effect of background, $F(1, 260)$ = 1.22, $p$ = .270, nor the interaction between background and target race, $F(1,260)$ = 1.95, $p$ = .16 reached significance.

Turning to treatment judgments, we again conducted a mixed-model ANOVA with participant background (public/MHP) as a between-subjects factor and target race as a within-subjects factor. Results of this analysis produced a significant main effect of target race, $F(1, 260)$ = 10.90, $p$ = .001, $\eta_p^2$ = .040 and a significant interaction between target race and participant background, $F(1, 260)$ = 6.50, $p$ = .011, $\eta_p^2$ = .024. The main effect of participant background fell short of significance, $F(1, 260)$ = 3.29, $p$ = .071, $\eta_p^2$ = .013.

To decompose this interaction, we first conducted paired samples t-tests for treatment judgments for Black and White targets separated by participant background. Lay participants again judged Black targets ($M$ = 11.69, $SD$ = 7.91) to need less treatment than White targets

($M$ = 12.21, $SD$ = 7.93), $t$(151) = -4.09, $p$ < .001, 95% CI[-.77, -.27], $d$ = 0.33. However, MHPs did not differ in their treatment judgments between Black and White targets, $t$(109) = -.58, $p$ = .56, 95% CI[-.29, .16], $d$ = 0.06. Additional independent samples t-tests analyzed the effect of background (public/MHP) on treatment for Black and White targets separately. These tests found that although members of the public ($M$ = 11.69, $SD$ = 7.01) judged Black targets to require significantly less treatment than MHPs ($M$ = 13.57, $SD$ = 6.51; $t$[255.58] = -2.11, $p$ = .036, 95% CI [-3.64, -.12], $d$ = 0.26), judgments for White targets did not significantly differ between lay participants and MHPs $t$(257.91) = -1.63, $p$ = .11, 95% CI[-3.15, .30], $d$ = 0.20.

As outlined in our pre-registration, using the MEMORE macro (Montoya & Hayes, 2017), we tested whether target race had an indirect effect on treatment judgments via racial biases in psychological distress. Mediation analyses were conducted separately for lay participants and MHPs. Among lay participants, race's effect on treatment outcomes was mediated by biased judgments of psychological distress, $b$ = -.37 $SE$ = .09, 95% CI [-.54, -.13]. However, analysis among MHPs revealed that target race's indirect effect on treatment recommendations via distress judgments did not reach significance, $b$ = -.14 $SE$ = .07, 95% CI [-.28, .01].

Suspecting that power prevented the pre-registered mediation analysis from achieving significance, we performed an exploratory mediation analysis on the full sample of MHPs (i.e., including MHPs that failed the study's attention check; n = 39). Before conducting this analysis, we first tested whether responses differed between MHPs that passed v. failed the attention check and whether attention check responses interacted with judgments of distress and treatment. Results of these analyses revealed that MHPs provided similar responses whether they passed or failed the attention check, $F$s< 1.00, $p$s>.36 and attention checks did not interact with effects of target race, $F$s <1.00, $p$s>.75. Bolstered by these results, we tested whether distress judgments mediated race's effect on treatment judgments for the full sample of MHPs (Fig 3). Results of this analysis provided evidence that among MHPs, racial biases in distress mediated target race's effects on treatment judgments, $b$ = -.17 $SE$ = .07, 95% CI [-.32, -.04].

## Discussion

These results provide further evidence that the public believes psychopathology harms Black individuals less than White individuals and, consequently, believe Black individuals have weaker treatment needs than White individuals. Critically, the current study provides suggestive evidence that these biases extend to practicing MHPs. Like members of the lay public,

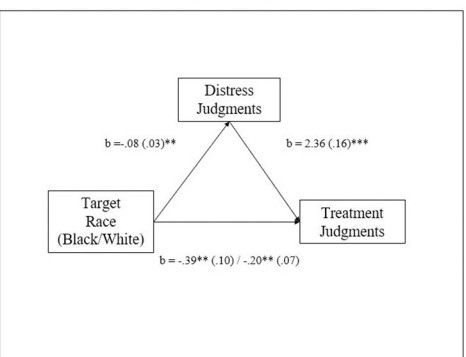
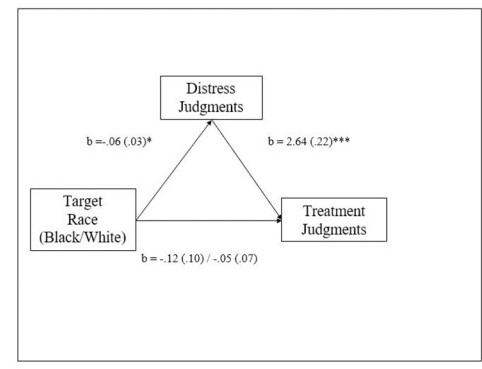

**Fig 3. Study 3 exploratory test of the effect of target race on treatment judgments is mediated by distress judgments.** Lay participants depicted in left panel. Mental health professionals depicted in right panel. Full sample analysis. * $p$≤.05, ** $p$≤.01, *** $p$≤.001.

MHPs judged Black targets to experience less psychopathology-related distress than White targets and—when analyzing data from all available MHPs—these biased judgments mediated target race's effect on treatment judgments. Although the direct effect of target race was not significant, exploratory analysis of all sampled MHPs revealed that race operated through racially-biased distress judgments to inform treatment recommendations. This exploratory analysis provides evidence that racial biases in judged distress generalize to MHPs and indirectly affect treatment judgments.

## General discussion

Recognizing distress is critical to mental healthcare. The mental health needs of Black Americans are particularly pertinent when considering the tolls of the COVID-19 pandemic and ongoing violence and harassment directed at Black Americans [3, 5]. Yet, at a time when Black Americans face multiple mental health crises, it appears that many in society—including some working in mental healthcare—paradoxically believe psychopathology hurts Black individuals less than White individuals. Evidence from four studies provide consistent evidence for this racial bias in mental health judgments. Lay participants and MHPs similarly believed that mental illness was less distressing to Black individuals than White individuals. Moreover, among lay participants, biased distress judgments mediated similar racial biases in care recommendations for Black relative to White individuals. Among MHPs, racially-biased distress judgments indirectly informed judgments of treatment needs. Although care should be taken when interpreting effects among MHPs, exploratory analysis of the complete sample suggests that racial biases in distress extend to practicing clinicians and indirectly influence treatment judgments for Black and White individuals.

### Implications

The current results make several key contributions. First, racially-biased judgments of psychological distress bear directly on the mental health crisis facing Black Americans. At a time when suicide rates and mental health needs are rising among Black Americans [34], portions of the public and MHPs paradoxically believe that psychopathology hurts Black individuals less than White individuals. Racial bias in distress judgments is directly relevant to the mental health challenges faced by Black Americans.

Second, racial biases in distress judgments represent a novel theoretical mechanism for understanding racial disparities in mental healthcare. Because judgments of distress are essential to every step of the treatment [29], racial disparities in psychological distress judgments stand to have numerous negative effects across the treatment process (e.g., assessment, symptom tracking, rapport building, care quality). Racial biases in distress judgments may contaminate numerous stages of mental healthcare.

Third, the minimization of Black individuals' psychological distress and treatment needs advances research on pain biases to chronic and extreme mental health conditions. These findings illustrate that race's effect on judgments of social hurts are not relegated to common everyday slights, but extend to judgments of highly distressing experiences and crucial mental health needs. By documenting race's effect on mental health judgments, these results advance research on biases in pain judgments generally and research on social pain biases specifically.

Fourth, racially-biased mental health judgments may affect multiple life contexts. At work and in school, authorities often have wide latitude for providing time off, excused absences, and deadline extensions. Widely held beliefs that minimize the psychological distress and mental health needs for Black individuals stand to undermine accommodations and support in numerous interpersonal contexts.

Finally, racial biases in mental health and treatment needs have negative implications for interventions and programming to ameliorate racial disparities in mental health. Perceived need fuels funding priorities in both the public and private sector [35]. Consequently, racial biases that undermine conceptions of the psychological distress and mental health needs of Black Americans may negatively affect government and philanthropic resources directed toward Black mental health.

## Limitations and future directions

Limitations offer avenues for future research. Although the current research offers compelling initial evidence for a race-based bias in psychological distress and consequent treatment judgments, caution should be exercised when extrapolating these experimental results to the population writ large and the diverse field of mental healthcare professionals. Among laypeople, although the consistent replication of distress and treatment judgments provides evidence of the generality of this bias, replication among a nationally representative sample of American adults would provide even greater support for the durability of these effects. Among clinicians, more detailed replication, particularly with larger and more diverse samples with ecologically richer stimuli would provide greater understanding of the effect of client race on MHPs' judgments of distress and treatment. First, larger samples affording greater statistical power will provide greater evidence of the reproducibility of Study 3's effects among MHPs. Second, future research should test the role of clinician race and training orientation to potentially moderate distress and treatment judgments. Although there is some evidence from research on physical and social pain judgments that Black and White individuals often show similar biases [10, 14], presumably because of the nominally positive nature of stereotypes related to Black strength and resilience, it is not clear whether these effects generalize to Black clinicians and those trained in more culturally competent approaches to mental health [36–38]. Future research should test for moderators among MHPs. Third, researchers might test how client race shapes distress judgments in more ecologically valid contexts. For example, guided by methods for testing social accuracy in emotion perception [39, 40], stimuli could be created in which actual Black and White clients report their distress before and during structured clinical interviews. Segments of these interviews could then be presented to novel clinician participants tasked with judging clients' distress at focal interview moments. To the extent that Black and White clients have moments of low and high distress and these experiences can be matched across target client race (e.g., segment trials in which Black and White clients indicate feeling a distress level of "5" on 7-point Likert scale), this procedure would allow researchers to manipulate target race while holding client distress and personal characteristics constant (see also [41] for rigorous procedures using computer generated stimuli). Additionally, such a procedure would allow researchers to not only test whether MHPs *minimize* Black clients' distress relative to White clients' distress, it would also afford tests of whether they *underestimate* Black clients' distress relative to the clients' actual emotions. Although resource and labor intensive, replicating Study 3's effects using such ecologically rich stimuli would provide even stronger evidence that race biases clinicians' judgments of Black clients' distress. In sum, although the current results offer initial evidence that laypeople and MHPs minimize the distress of psychopathology for Black relative to White individuals, care should be taken when extrapolating these findings to the broad field of mental health. Future research would provide greater support for these findings by testing effects with bigger and more diverse samples and with even richer experimental stimuli.

Just as pressing, future research must also investigate mechanisms to eliminate racial biases in mental health and treatment needs. Considering evidence that mindfulness and

perspective-taking sometimes spotlight the harms of discrimination and racial inequity [42, 43], strategies geared at improving perspective-taking and mindfulness may mitigate biased distress and treatment judgments.

Another limitation of the current work is its exclusive use of male target individuals. Although there is evidence that racial biases in social pain judgments occur for Black female and Black male target individuals [10] and clinicians under-diagnose certain forms of psycho-pathology (e.g., eating disorders) for Black female relative to White female clients [44], other social cognitive work highlights that race is sometimes gendered, producing unique biases toward Black female targets [45]. Future research should test whether target race and gender produce independent or interactive effects on judgments of psychological distress and treatment recommendations.

Future research should also test whether racially-biased distress judgments affect the therapy strategies MHPs provide Black clients. There is evidence that clinicians are less likely to use cognitive change strategies with Black versus White clients [46]. To the extent that cognitive change strategies are critical to care, providing less of this "ingredient" may dilute therapy's effectiveness for Black clients. Future research should examine whether clinicians' racial biases in distress judgments shape therapy decisions, retention, and care outcomes.

Finally, future research might also explore how biases in psychological distress are shaped by other group identities. For example, other social groups are judged as tough and insensitive to pain (e.g., low-SES individuals [47]), have their emotional capacities dehumanized (e.g., larger-bodied individuals [48]), and feel their social hurts are invalidated (e.g., East/South-East Asian, Asian-American, Indigenous, and Latine individuals [49]). Future research should test whether biases in mental health and treatment needs extend to individuals from diverse social backgrounds.

## Conclusions

At a time when Black Americans' mental health needs are pressing, the current work provides consistent evidence for a racial bias in mental health and treatment judgments. Members of the public and practicing mental healthcare providers believed psychopathology harms Black individuals less than White individuals and these distress biases informed treatment judgments. These biases represent a social-cognitive obstacle to achieving mental healthcare equity and require urgent address to meet the critical needs of Black Americans.

## Supporting information

**S1 File.**
(DOCX)

## Author Contributions

**Conceptualization:** Jonathan W. Kunstman, Tade Ogungbadero, Jason C. Deska, Kurt Hugenberg.

**Data curation:** Jonathan W. Kunstman.

**Formal analysis:** Jonathan W. Kunstman.

**Funding acquisition:** Kurt Hugenberg.

**Methodology:** Jonathan W. Kunstman.

**Project administration:** Jonathan W. Kunstman.

**Visualization:** Tade Ogungbadero.

**Writing – original draft:** Jonathan W. Kunstman, Tade Ogungbadero.

**Writing – review & editing:** Jonathan W. Kunstman, Tade Ogungbadero, Jason C. Deska, Michael J. Bernstein, April R. Smith, Kurt Hugenberg.

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
