## [Decision Letter · Decision Letter 0]

31 Aug 2023

PONE-D-23-11067Race-Based Biases in Psychological Distress and Treatment JudgmentsPLOS ONE

Dear Dr. Kunstman,

Thank you for submitting your manuscript to PLOS ONE. After careful consideration, we feel that it has merit but does not fully meet PLOS ONE’s publication criteria as it currently stands. Therefore, we invite you to submit a revised version of the manuscript that addresses the points raised during the review process.

Thank you for the submission of your manuscript. In your revision, please attend to the comments of Reviewer 1 in addition to these points below: -Please provide citations to support the following statement in the introduction: "Consequently, Black Americans’ mental health suffers at significantly greater rates than White Americans." -Please briefly describe the Chicago Face Database and describe how the faces used for the studies were selected. -Please briefly describe MTurk and CloudResearch in the manuscript.  -Did the authors use any strategies to account for/rule out bot responses in the crowd sourced samples?  -Please give the exact wording for the "mental health professional" screening item(s) used in each study where this question was asked.  -I was surprised that the authors did not appear to account for participants' race in their analyses. Can they explain why this was not done?

We look forward to receiving your revised manuscript.

Kind regards,

Emily Lund

Academic Editor

PLOS ONE

Journal Requirements:

2. Your abstract cannot contain citations. Please only include citations in the body text of the manuscript, and ensure that they remain in ascending numerical order on first mention.

Additional Editor Comments:

Thank you for the submission of your manuscript. In your revision, please attend to the comments of Reviewer 1 in addition to these points below:

-Please provide citations to support the following statement in the introduction: "Consequently, Black Americans’ mental health suffers at significantly greater rates than White Americans."

-Please briefly describe the Chicago Face Database and describe how the faces used for the studies were selected.

-Please briefly describe MTurk and CloudResearch in the manuscript.

-Did the authors use any strategies to account for/rule out bot responses in the crowd sourced samples?

-Please give the exact wording for the "mental health professional" screening item(s) used in each study where this question was asked.

-I was surprised that the authors did not appear to account for participants' race in their analyses. Can they explain why this was not done?

Reviewers' comments:

Reviewer's Responses to Questions

**Comments to the Author**

1. Is the manuscript technically sound, and do the data support the conclusions?

Reviewer #1: Yes

2. Has the statistical analysis been performed appropriately and rigorously? 

Reviewer #1: Yes

3. Have the authors made all data underlying the findings in their manuscript fully available?

Reviewer #1: Yes

4. Is the manuscript presented in an intelligible fashion and written in standard English?

Reviewer #1: Yes

5. Review Comments to the Author

Reviewer #1: The authors present four studies examining racial bias related to the perceived impact of psychopathology (i.e., how distressing it will be for Black vs. White individuals) as well as subsequent treatment needs. I applaud the authors for including pre-registered studies and also examining this bias in both lay individuals and in mental healthcare professionals.

In the introduction, when addressing the contributions of the current work I think that the paragraphs beginning “Fourth, beyond mental healthcare” and “Finally, racial biases in mental health” should be reserved solely for the discussion section (where very similar paragraphs already exist). The three preceding paragraphs are the true (and rich) contributions to the literature, these latter two paragraphs really speak more to future directions and longer-term implications of this work.

In the final iterations of this manuscript, I would like to see the authors spend a bit more space discussing the limitations/caution needed in interpreting Study 3 results, and perhaps direct recommendations for future research that would remedy these limitations and more firmly support the idea that this bias is consistently found in MHPs as well as the general public.

6. PLOS authors have the option to publish the peer review history of their article (what does this mean?). If published, this will include your full peer review and any attached files.

Reviewer #1: No

---

## [Author Response · Author response to Decision Letter 0]

13 Sep 2023

Dear Dr. Lund, 

Thank you for your consideration of our manuscript entitled ‘Race-Based Biases in Psychological Distress and Treatment Judgments.’ We were excited for the opportunity to revise and resubmit the manuscript. We appreciated the thoughtful and overall positive response provided by you and Reviewer 1. Guided by this constructive feedback, we have worked diligently to improve the paper. 

Below we summarize our responses and modifications to the manuscript. We first respond to your comments and then the points of Reviewer 1. For clarity, revision items from the review have been italicized. 

AE Comments

-Please provide citations to support the following statement in the introduction: "Consequently, Black Americans’ mental health suffers at significantly greater rates than White Americans."

Citations have been added (i.e., DeVylder et al., 2020; Eboigbe et al., 2023; Novacek et al., 2020). 

-Please briefly describe the Chicago Face Database and describe how the faces used for the studies were selected.

Additional information related to the CFD has been added to the introduction to Study 1 (pg. 6). Specifically, we now note:

The CFD is a frequently used and well validated stimulus set of adult target individuals from a variety of racial and ethnic backgrounds. Using norming data associated with face stimuli, we identified Black and White target individuals whose racial identity was easily identifiable while holding constant other facial features that might affect distress judgments (e.g., dominance, attractiveness, baby-facedness). Thus, using normed CFD stimuli allowed us to manipulate target race without introducing confounds related to facial structure and morphology.

-Please briefly describe MTurk and CloudResearch in the manuscript.

Descriptions of MTurk and CloudResearch have been added to pages 7 and 14, respectively. 

MTurk is an internet-based participant service that allows researchers to connect with individuals interested in participating in social science research in exchange for payment to Amazon Marketplace.

CloudResearch. Like MTurk, Cloudresearch is an online participant platform that allows interested individuals to participate in research in exchange for monetary compensation.

-Did the authors use any strategies to account for/rule out bot responses in the crowd sourced samples?

We now better highlight that duplicate IP addresses were used as an exclusion criteria to mitigate bot responses. We further clarify that these decision rules were included in our study pre-registrations. See pages 7, 11, and 15. 

-Please give the exact wording for the "mental health professional" screening item(s) used in each study where this question was asked.

The exact wording of the mental health screener has been added to page 14. The item read:

“From the list below, please indicate which field best represents your profession.” To be eligible to advance to the distress and treatment task, participants needed to select the option “Mental Health: counselors, clinicians, therapists, psychologists, psychiatrists, social workers, etc.” 

-I was surprised that the authors did not appear to account for participants' race in their analyses. Can they explain why this was not done?

We did not take up the issue of participant race in analyses because research in both physical and social pain judgments frequently finds that participants of color generally and Black participants specifically frequently show comparable biases to White participants (e.g., Deska et al., 2020b; Trawalter et al., 2012). Indeed, some work finds that Black participants sometimes show slightly larger biases than White participants (see Deska et al., 2020a; Study 3), presumably because these effects are associated with nominally positive stereotypes about Black strength and resilience (values that carry ingroup significance for Black Americans’ past and present struggles for civil rights in the U.S. cultural context; Baldwin, 1998; Carson & Shepard, 2001). We now more directly highlight this rationale on pg 22 of the General Discussion and also make the point—related to Reviewer 1’s call for caution---to recommend that future research take up the role of MHP race in tests of replication and moderation of these effects. 

The manuscripts formatting and style has been updated in keeping with PLOS One’s guides. 

R1 Comments

In the introduction, when addressing the contributions of the current work I think that the paragraphs beginning “Fourth, beyond mental healthcare” and “Finally, racial biases in mental health” should be reserved solely for the discussion section (where very similar paragraphs already exist). The three preceding paragraphs are the true (and rich) contributions to the literature, these latter two paragraphs really speak more to future directions and longer-term implications of this work.

In keeping with this recommendation, these paragraphs have been removed from the introduction and are highlighted exclusively in the General Discussion. 

In the final iterations of this manuscript, I would like to see the authors spend a bit more space discussing the limitations/caution needed in interpreting Study 3 results, and perhaps direct recommendations for future research that would remedy these limitations and more firmly support the idea that this bias is consistently found in MHPs as well as the general public.

Guided by this recommendation, we now explicitly open the limitations section by calling for caution to avoid overgeneralizing these results. On pages 22-23, we then outline three areas that would provide additional evidence for the generalization of Study 3’s findings, particularly as they relate responses among MHPs. In brief, those three recommendations are 1) larger samples to provide greater statistical power to test client race’s effect on distress and treatment judgments, 2) enhancing the diversity of MHPs and their associated training orientations, particularly in regard to culturally competent therapy, and 3) describe using a social accuracy paradigm that could used with clinician participants to provide a more ecologically rich test of the hypothesis that client race biases distress and treatment biases. 

For convenience, the complete version of this text in included below: 

Limitations offer avenues for future research. Although the current research offers compelling initial evidence for a race-based bias in psychological distress and consequent treatment judgments, caution should be exercised when extrapolating these experimental results to the population writ large and the diverse field of mental healthcare professionals. Among laypeople, although the consistent replication of distress and treatment judgments provides evidence of the generality of this bias, replication among a nationally representative sample of American adults would provide even greater support for the durability of these effects. Among clinicians, more detailed replication, particularly with larger and more diverse samples with ecologically richer stimuli would provide greater understanding of the effect of client race on MHPs’ judgments of distress and treatment. First, larger samples affording greater statistical power will provide greater evidence of reproducibility of Study 3’s effects among MHPs. Second, future research should test the role of clinician race and training orientation to potentially moderate distress and treatment judgments. Although there is some evidence from research on physical and social pain judgments that Black and White individuals often show similar biases10,14, presumably because of the nominally positive nature of stereotypes related to Black strength and resilience, it is not clear whether these effects generalize to Black clinicians and those trained in more culturally competent approaches to mental health36-38. Future research should test for moderators among MHPs. Third, researchers might test how client race shapes distress judgments in more ecologically valid contexts. For example, guided by methods for testing social accuracy in emotion perception39,40, stimuli could be created in which actual Black and White clients report their distress before and during structured clinical interviews. Segments of these interviews could then be presented to novel clinician participants tasked with judging clients’ distress at focal interview moments. To the extent that Black and White clients have moments of low and high distress and these experiences can be matched across target client race (e.g., segment trials in which Black and White clients indicate feeling a distress level of “5” on 7-point Likert scale), this procedure would allow researchers to manipulate target race while holding client distress and personal characteristics constant (see also41 for rigorous procedures using computer generated stimuli). Additionally, such a procedure would allow researchers to not only test whether MHPs minimize Black clients’ distress relative to White clients’ distress, it would also afford tests of whether they underestimate Black clients’ distress relative to the clients’ actual emotions. Although resource and labor intensive, replicating Study 3’s effects using such ecologically rich stimuli would provide even stronger evidence that race biases clinicians’ judgments of Black clients’ distress. In sum, although the current results offer initial evidence that laypeople and MHPs minimize the distress of psychopathology for Black relative to White individuals, care should be taken when extrapolating these findings to the broad field of mental health. Future research would provide greater support for these findings by testing effects with bigger and more diverse samples and with even richer experimental stimuli.

In conclusion, we thank the review team for the thoughtful evaluation of our work. Guided by your comments, we’ve worked carefully to update the manuscript. We hope these modifications have elevated the paper to the level necessary for publication at PLOS One.

---

## [Editor Report · Decision Letter 1]

5 Oct 2023

Race-Based Biases in Psychological Distress and Treatment Judgments

PONE-D-23-11067R1

Dear Dr. Kunstman,

We’re pleased to inform you that your manuscript has been judged scientifically suitable for publication and will be formally accepted for publication once it meets all outstanding technical requirements.

Kind regards,

Emily Lund

Academic Editor

PLOS ONE
---

## [Editor Report · Acceptance letter]

10 Oct 2023

PONE-D-23-11067R1 

Race-Based Biases in Psychological Distress and Treatment Judgments 

Dear Dr. Kunstman:

I'm pleased to inform you that your manuscript has been deemed suitable for publication in PLOS ONE. Congratulations! Your manuscript is now with our production department. 

Kind regards, 

on behalf of

Dr. Emily Lund 

Academic Editor

PLOS ONE